# Investigation of Strategies to Block Downstream Effectors of AT1R-Mediated Signalling to Prevent Aneurysm Formation in Marfan Syndrome

**DOI:** 10.3390/ijms25095025

**Published:** 2024-05-04

**Authors:** Irene Valdivia Callejon, Lucia Buccioli, Jarl Bastianen, Jolien Schippers, Aline Verstraeten, Ilse Luyckx, Silke Peeters, A. H. Jan Danser, Roland R. J. Van Kimmenade, Josephina Meester, Bart Loeys

**Affiliations:** 1Centre of Medical Genetics, Antwerp University Hospital, University of Antwerp, 2650 Antwerp, Belgium; irene.valdiviacallejon@uantwerpen.be (I.V.C.); lucia.buccioli@uantwerpen.be (L.B.); jarl.bastianen@uantwerpen.be (J.B.); jolien.schippers@uantwerpen.be (J.S.); aline.verstraeten@uantwerpen.be (A.V.); ilse.luyckx@uantwerpen.be (I.L.); silke.peeters@uantwerpen.be (S.P.);; 2Department of Human Genetics, Radboud University Medical Center, 6525 GA Nijmegen, The Netherlands; 3Department of Internal Medicine, Erasmus Medical Center, 3015 GD Rotterdam, The Netherlands; a.danser@erasmusmc.nl; 4Department of Cardiology, Radboud University Medical Center, 6525 GA Nijmegen, The Netherlands; roland.vankimmenade@radboudumc.nl

**Keywords:** Marfan syndrome, AT1R antagonism, β-arrestin, aortic aneurysm

## Abstract

Cardiovascular outcome in Marfan syndrome (MFS) patients most prominently depends on aortic aneurysm progression with subsequent aortic dissection. Angiotensin II receptor blockers (ARBs) prevent aneurysm formation in MFS mouse models. In patients, ARBs only slow down aortic dilation. Downstream signalling from the angiotensin II type 1 receptor (AT1R) is mediated by G proteins and β-arrestin recruitment. AT1R also interacts with the monocyte chemoattractant protein-1 (MCP-1) receptor, resulting in inflammation. In this study, we explore the targeting of β-arrestin signalling in MFS mice by administering TRV027. Furthermore, because high doses of the ARB losartan, which has been proven beneficial in MFS, cannot be achieved in humans, we investigate a potential additive effect by combining lower concentrations of losartan (25 mg/kg/day and 5 mg/kg/day) with barbadin, a β-arrestin blocker, and DMX20, a C-C chemokine receptor type 2 (CCR2) blocker. A high dose of losartan (50 mg/kg/day) slowed down aneurysm progression compared to untreated MFS mice (1.73 ± 0.12 vs. 1.96 ± 0.08 mm, *p* = 0.0033). TRV027, the combination of barbadin with losartan (25 mg/kg/day), and DMX-200 (90 mg/kg/day) with a low dose of losartan (5 mg/kg/day) did not show a significant beneficial effect. Our results confirm that while losartan effectively halts aneurysm formation in *Fbn1^C1041G/+^* MFS mice, neither TRV027 alone nor any of the other compounds combined with lower doses of losartan demonstrate a notable impact on aneurysm advancement. It appears that complete blockade of AT1R function, achieved by administrating a high dosage of losartan, may be necessary for inhibiting aneurysm progression in MFS.

## 1. Introduction

Marfan syndrome (MFS) is a heritable disorder characterized by abnormalities in the connective tissues, primarily affecting the ocular, skeletal, and cardiovascular systems [1]. It is caused by pathogenic variants in the *FBN1* gene and follows an autosomal dominant inheritance pattern, although a very rare number of MFS families have been reported to harbour autosomal recessive variants [2]. While MFS typically segregates within families, around 30% of cases are caused by de novo variants [1,3].

Amongst all clinical features, the cardiovascular manifestations of MFS contribute most significantly to the observed morbidity and mortality. The primary cardiovascular characteristic is the enlargement of the aortic root, specifically at the sinus of Valsalva, a feature present in the majority of patients. There can be considerable variability in the rate of this dilation among affected individuals. As a result of the aortic root enlargement, aortic valve insufficiency with subsequent regurgitation can develop. Dilation may progress to an aneurysm, potentially leading to a type A dissection or rupture. A small percentage of patients (10–20%) show dilation of the descending and abdominal aorta, which can result in a type B dissection [3].

At the mechanistic level, an increase in transforming growth factor beta (TGF-β) pathway activity has been observed in aortic wall samples from individuals with syndromic thoracic aortic aneurysm (TAA), as well as in mouse models of MFS [3,4,5]. The *FBN1* gene encodes for the fibrillin-1 protein, a structural macromolecule that forms microfibrils through polymerisation. These microfibrils are found in the connective tissue, contributing to the creation of tissue-specific frameworks and serving as reservoirs for growth factors, such as TGF-β. Due to pathogenic variants in the *FBN1* gene, these fibrillin microfibrils fail to adequately sequester TGF-β within latent TGF-β binding complexes connected to fibrillin-1 [6]. This disruption is believed to lead to the observed dysregulation of the TGF-β pathway. Since this pathway regulates several processes in cells, its dysregulation may eventually contribute to other molecular features described in the aortic walls of TAA samples, including the phenotypic switching of vascular smooth muscle cells (VSMCs), endothelial dysfunction, changes in the extracellular matrix (ECM), and inflammation [5,7,8]. 

Despite recent advancements in our understanding of TAA pathomechanisms, the available pharmacological options remain quite limited. β-blockers are widely acknowledged as the standard of care required to decrease the aortic dilation rate by reducing blood pressure. Nevertheless, the trials conducted so far do not demonstrate a full arrest of aortic growth, highlighting the necessity of additional studies to determine the effectiveness of β-blockers in this context [9]. The use of more cardiac-specific β-blockers might improve tolerability, but certain individuals may still experience intolerance to medications within this class of drugs. Hence, for this group of patients, the use of angiotensin II type 1 receptor blockers (ARB) can be considered an alternative or adjunctive therapy [10]. This class of medication has been used for treating renal disorders characterised by elevated levels of TGFβ, providing additional evidence for their use in other conditions associated with excessive TGFβ signalling [11]. While the use of the ARB losartan selectively antagonises the angiotensin II type 1 signalling only, debate about the contribution of angiotensin II type 1 receptor (AT1R) and angiotensin II type 2 receptor (AT2R) to aortic aneurysm is still ongoing. It is suggested that the two receptors exert opposite effects: the former appears to contribute to pathogenesis, while the latter is suggested to have a protective effect [4]. 

Despite the very encouraging results from the preclinical losartan study in a murine MFS model [12], the ARB losartan did not show superior effects compared to standard therapy (i.e., β-blockade) in a human comparative trial [13]. The largest trial showed that a high dose of β-blockers (up to four-fold of the normal dose) performed equally well as a low dose of losartan [13,14]. A meta-analysis also suggested that treatment with β-blockers and losartan could have an additive effect, although the study did not have enough power to prove it [10]. Furthermore, subsequent trials have shown that higher doses of sartans perform better (personal communication: Dr. Harry C. Dietz). The latter was also shown in the MFS mouse model: a high dose of losartan performed much better than low or medium doses of losartan. The human equivalent dose of this high oral dose of losartan would be 4–8 mg/kg/day, which is much higher than the doses commonly used in a human setting. Currently used doses in MFS patients are up to 2 mg/kg/day in children and 100–150 mg per day (1.4–2.1 mg/kg/day) in adults. These higher doses in humans may result in more side effects.

Although current therapies targeting AT1R do not completely prevent aortic dilation and dissection, the involvement of AT1R in TAA seems to be evident. Exploring new approaches for targeting AT1R remains a promising strategy to improve treatment for TAA. In particular, AT1R blocking strategies that would reinforce the use of lower doses of losartan would be very attractive. AT1R signalling involves two different components: G protein activation, which promotes vasoconstriction, and β-arrestin recruitment, which leads to enhanced cardiomyocyte contractility and cell survival. Classical ARBs, such as losartan, inhibit both pathways [15]. Nevertheless, emerging evidence indicates that biased ligands can selectively target these pathways independently, engaging specific subsets of the normal signalling repertoire of the receptor, particularly the β-arrestin pathway. However, the precise effects of such selectivity are yet to be fully understood [16]. 

TRV027 is a β-arrestin-biased AT1R ligand that competitively inhibits G-protein-dependent signalling while promoting β-arrestin recruitment. In rats, it has been shown to reduce arterial pressure similarly to an ARB, but unlike an ARB, it also enhances cardiac contractility and maintains stroke volume [17]. In both in vitro and in vivo settings, TRV027 has been demonstrated to have antiapoptotic and inotropic effects in wild-type (WT) mice but not in β-arrestin knock-out mice [17]. In a dog model of acute heart failure, TRV027 has acted as a potent and balanced vasodilator, improving cardiac output and offering renal benefits [18]. In a *ApoE^−/−^* mouse model in which aortic aneurysms were induced by angiotensin II type 1 (Ang II), the co-infusion of Ang II with TRV027 prevented the formation of these aneurysms. TRV027 halted some detrimental effects such as aortic dilation, asymmetric wall thickening, inflammation, vascular fibrosis, elastolysis, and the seepage of blood [19]. These findings suggest that TRV027 could offer therapeutic benefits by blocking G-protein-mediated vasoconstriction and other damaging processes while simultaneously promoting cardio- and reno-protective β-arrestin-mediated signalling [18,19,20]. 

Although the above-mentioned studies have shown that β-arrestin-biased stimulation may result in positive cardiovascular effects, the literature also suggests positive effects following its deletion. A murine MFS model with genetic β-arrestin (βarr2) deletion, i.e., *Fbn1*^C1041G/+^/βarr2^−/−^ mice, exhibited delayed aortic root dilation compared to *Fbn1*^C1041G/+^ mice. Moreover, the aortas of *Fbn1*^C1041G/+^/*βarr2*^−/−^ mice showed reduced mRNA and protein expression of key mediators involved in TAA formation, including matrix metalloproteinase (MMP)-2 and -9, along with decreased activation of ERK1/2, compared to *Fbn1*^C1041G/+^ mice [21]. Using primary aortic root smooth muscle cells where β-arrestin was targeted through small interfering RNAs, it was demonstrated that the induction of MMP-2 and -9 expression by Ang II relies on βarr2. This pathway involves the activation of ERK1/2 and transactivation of epidermal growth factor receptor (EGFR). These results suggest that β-arrestin may play a non-canonical role in TAA formation in MFS by regulating the ERK1/2-dependent expression of pro-aneurysmal genes and proteins downstream of the AT1R [21]. β-arrestin deficiency has also been shown to attenuate abdominal aortic aneurysm (AAA) in an *ApoE^−/−^* mouse model in which AAA was induced by Ang II infusion [22].

Another emerging important player in TAA development is vascular inflammation [23,24,25]. Tieu et al. showed that Ang II infusion in C57BL/6J mice led to the production of monocyte chemoattractant protein 1 (MCP-1). The activation of MCP-1-mediated pathways resulted in the progression of aneurysm growth and dissection following Ang II infusion, while MCP-1 knock-out mice showed delayed aneurysmal growth [26]. Previous studies have also suggested an interplay between the AT1R and the receptor for MCP-1, known as a C-C chemokine receptor type 2 (CCR2). Functional heteromers of the AT1R and CCR2 resulted in the CCR2–G protein coupling, sensitive to AT1R activation, as well as to apparent enhanced β-arrestin recruitment with agonist co-stimulation. Moreover, in a rat model of chronic renal disease, it was observed that combined treatment with AT1R and CCR2 selective inhibitors was synergistically beneficial [27].

Thus, the aim of this study is to explore alternative strategies for targeting AT1R-mediated signalling that could be beneficial in MFS. To achieve this, we first investigate the effect of targeting β-arrestin downstream signalling by administering TRV027, which promotes β-arrestin recruitment. Secondly, we evaluate the combined effects of the biased AT1R ligands barbadin and DMX200 with low doses of the ARB losartan as a potential strategy to enhance effectiveness while potentially minimizing side effects (Figure 1).

## 2. Results

### 2.1. TRV027 Infusion Does Not Show Any Beneficial Effect on Ascending Aorta and Aortic Root Dilation in a MFS Mouse Model

Based on its previously reported positive effect on blood pressure and aortic distensibility [17,18,19], we hypothesised that TRV027 infusion in MFS mice may have a positive effect in preventing aortic dilation. Small osmotic pumps were subcutaneously implanted in male littermate MFS (*Fbn1^C1041G/+^*) and WT mice, achieving a TRV027 infusion rate of 10 µg/kg/min. A control group was also included, in which mice were infused with saline solution. At 4 weeks of age, just before the start of the treatment, MFS mice already showed dilation of the aortic root compared to WT (Figure 2), while no dilation of the ascending aorta was detected (Appendix A). At 8 and 12 weeks of age, after 4 and 8 weeks of treatment, respectively, no significant differences were observed on aortic root and ascending aorta diameters between MFS and WT mice infused with TRV027 and saline solution. MFS mice from both groups showed significant dilation of the aortic root at 8 and 12 weeks compared to their WT counterparts (at 8 weeks, *p* < 0.0001 for TRV027-treated group and *p* = 0.0016 for the saline control group; at 12 weeks, *p* < 0.0001 for both TRV027 and the control group) (Figure 2). These results suggest that TRV027 treatment has no impact on the aortic root or the ascending aorta enlargement of MFS or WT mice.

### 2.2. Combined Treatment of Barbadin and Losartan Does Not Show a Significant Effect on Ascending Aorta and Aortic Root Diameter Compared to Losartan Alone

Based on the previous observation that the knock-out of β-arrestin-2 improved aortic dilation in the MFS *Fbn1*^C1041G/+^ mouse model, we hypothesised that combined treatment with the biased β-arrestin blocker barbadin, alongside a lower dose of losartan, may have similar effects on the prevention of aortic dilation to a high dose of losartan by itself. This idea raises the possibility of achieving superior efficacy without facing side effects associated with high doses of losartan in humans. To test the hypothesis, we gave a combined treatment of both losartan (25 mg/kg/day) and barbadin (0.9 mg/kg/day) to WT and MFS mice and compared them with MFS and WT treated with losartan alone (25 mg/kg/day). The treatment started at 4 weeks of age and continued until 12 weeks of age. Our results showed that aortic root diameters were already increased in MFS mice from 4 weeks of age, just before the start of the treatment (Figure 3 and Appendix A). When comparing both treated groups, combined treatment with both losartan (25 mg/kg/day) and barbadin (0.9 mg/kg/day) did not show any significant improvement in the aortic root dilation over time compared to using losartan (25 mg/kg/day) alone (Figure 3 and Appendix A). Both treated MFS mice groups showed significant dilation compared to their WT counterparts at 8 and 12 weeks (at 8 weeks, *p* < 0.0001 for barbadin- and losartan-treated mice and *p* = 0.061 for losartan-treated mice; at 12 weeks, *p* = 0.0087 for barbadin- and losartan-treated mice and *p* < 0.0001 for losartan-treated mice) (Figure 3 and Appendix A). These findings suggest that an additive treatment with barbadin does not exert any effect on the aortic phenotype of MFS mice. It is noteworthy that this experiment does not include a group treated with barbadin alone. Although we observed no synergistic effect from the combined treatment of barbadin and losartan, it cannot be concluded what the impact of barbadin alone is on aortic dilatation in MFS mice. Additional interactions with losartan could potentially influence the effect of barbadin when administered together. A subtle decrease in aortic diameter in both treated WT and MFS mice was observed at 8 and 12 weeks compared with their respective genotypes in the untreated group (Figure 3). However, these differences did not reach statistical significance in either case. Since the same decrease was observed in both treated groups, this suggests that losartan alone may be responsible for this effect. This finding suggests that while losartan at a dose of 25 mg/kg/day has some beneficial effect on aortic growth in both MFS and WT mice, it is not enough to prevent dilation in MFS mice. As for the ascending aorta measurements, there was no evident dilation in MFS mice at the ages when diameters were measured (4, 8, and 12 weeks of age), and no distinction was observed between the two treated groups (Appendix A). Additionally, there were no differences in ascending aortic diameter between treated and untreated mice (Appendix A).

### 2.3. Combined Treatment of DMX-200 and Low Dose of Losartan Does Not Show Any Significant Effect on the Ascending Aorta and Aortic Root Diameter of MFS Mice Compared to Both Treated and Untreated WT Groups

Based on prior studies in the context of chronic kidney disease, we hypothesised that co-inhibition of AT1R and CCR2 may result in a decrease in β-arrestin recruitment but also highly affect CCR2–G protein signalling, possibly through allosteric modulation [27]. Another study has assessed the combinatory effect of DMX200 with an ARB in the context of COVID-19 to potentially achieve a synergistic anti-inflammatory effect [28]. We propose that combinatory blockage of AT1R and CCR2 may result in selective β-arrestin inhibition while also preventing inflammatory cascades, having to a positive effect on aneurysm formation. From these observations, we aimed to find if the combination of CCR2blocker DMX-200 with a low dose of losartan could produce comparable outcomes in averting aortic dilation as a high dose of losartan alone. Hence, WT and MFS mice received a combined treatment of a low dose of losartan (5 mg/kg/day) and DMX-200 (90 mg/kg/day) and were compared with a group treated with high dose of losartan alone (50 mg/kg/day). As shown in Figure 4, we observed that, at both 8 and 12 weeks of age, the MFS group treated with high dose of losartan showed no statistically significant difference with the treated WT group. This suggests that a high dose of losartan can prevent aneurysm development in our murine MFS model. Moreover, a reduction in aortic diameter was noted in the WT group upon treatment with a high dose of losartan at 8 and 12 weeks, as opposed to its untreated counterpart. This can also be observed when considering the growth rate of the aortic root diameter, with treatment with a high dose of losartan resulting in a lower growth rate in both WT and MFS mice compared to both untreated groups (Figure 5). On the other hand, the MFS group treated with low dose of losartan and DMX-200 showed a statistically significant difference (*p* < 0.0209) in aortic diameter when compared to the WT counterpart at 12 weeks. However, when compared to the MFS group treated with high dose of losartan, it was observed that the difference was not statistically significant. This might also suggest a trend in decreased aortic dilation in the group treated with losartan and DMX-200. It is important to note that DMX200 was not tested alone, and the effect of the low dose of losartan (5 mg/kg/day) is also unknown. We can conclude that combination of these two compounds at the tested doses does not exhibit any significant positive effect on aortic dilatation in MFS mice. However, we cannot determine the individual effects of DMX200 and the low dose of losartan, as a potential interaction between the two could be influencing the outcome when administered together.

## 3. Discussion

Despite very promising results in an MFS mouse model, human clinical studies pursuing AT1R blockade achieved less convincing results [13]. A meta-analysis of all ARB/β-blocker human studies showed that both ARBs and β-blockers slow down aortic root dilation to some extent but do not provide full rescue of the phenotype as in mice [10]. One of the possible explanations is that the doses of ARBs in the mouse trials were very high and, thus, might not have been tolerable in humans. AT1R activation triggers two different signalling components: G protein cascade, which promotes vasoconstriction, and β-arrestin recruitment, which leads to enhanced cardiomyocyte contractility and cell survival [29]. Interestingly, recent findings suggest that biased ligands can selectively target these pathways independently, engaging specific subsets of the normal signalling repertoire of the receptor, particularly the β-arrestin pathway. However, the precise effects of such selectivity remain not fully understood [16]. As such, we tested different pharmaceutical strategies aiming at the modulation of the AT1R/β-arrestin signalling pathway in the *Fbn1*^C1041G/+^ MFS mouse model. Existing data on the role of β-arrestin have been ambiguous. In this study, we explored if biased inhibition or stimulation plays a positive, negative, or neutral role in the AT1R effects. Furthermore, it has been shown that the MCP-1 -CCR2 pathway plays a role in aneurysm development [26]. Hence, this study also aimed to investigate if the combined inhibition of the AT1R and CCR2 may represent a therapeutic avenue for aortic aneurysm development. 

TRV027 has shown cardio- and reno-protective and antiapoptotic effects in different animal models [17,18,19]. At a dose of 10 μg/kg/day, TRV027 exhibits potent G protein inhibition with β-arrestin recruitment, resulting in lowered arterial pressure and increased cardiac contractility in rats [17]. Despite these promising outcomes, our findings indicate that TRV027 infusion at a rate of 10 μg/kg/day does not influence the aortic diameter in the MFS model *Fbn1^C1041G/+^.* This could suggest that solely inhibiting G-protein-dependent signalling is not sufficient to prevent TAA in MFS. The potential counteraction of any G protein blocking effect by β-arrestin recruitment, promoted by TRV027, could also contribute to this lack of effect on aortic root diameter in the MFS model. Additionally, the dose used in this study, while effective in producing cardiac-related benefits in rats, might not be sufficient to exert any observable effect on aortic dilation in MFS mice. Increasing the concentration of TRV027, however, presents challenges due to the limited solubility of TRV027 in water (300 mg/mL). Furthermore, other limitations include insufficient pharmacological data on the efficacy and stability of TRV027 administration via infusion with minipumps.

Our findings also indicate that barbadin, a known selective inhibitor of β-arrestin, did not confer any benefits in mitigating aortic aneurysm progression in *Fbn1*^C1041G/+^ mice. The anticipated positive outcome was based on previous studies demonstrating delayed aortic root dilation in *Fbn1*^C1041G/+^/*βarr2*^−/−^ mice, suggesting that inhibiting β-arrestin with a compound might result in a similar effect. Research about the effects of different concentrations of barbadin is limited but has shown that a dosage of 0.3 mg/kg/day is able to potentiate the effects of the weight-loss drug lorcaserin in male mice [30]. Nevertheless, our study found that, even with a higher dose, there is no impact on the aortic diameter of MFS mice. The observed lack of efficacy could potentially be attributed to the need for a more potent β-arrestin blockade, considering that gene knock-out represents a more robust intervention compared to receptor blocking using a compound. Higher doses of barbadin might be required to exert a therapeutic effect. However, such doses may deviate from clinically relevant levels (no data are currently available), raising concerns about the translatability of these findings to human patients. 

Losartan has already been reported to prevent aortic dilation in the MFS model *Fbn1^C1041G/+^* [12], and with our study, we confirmed the efficacy of a high dose of losartan (50 mg/kg/day). AT1R and CCR2 were reported to functionally interact, and combined treatment with the CCR2 inhibitor DMX-200 (30 mg/kg/day) and irbesartan, another AT1R antagonist, synergistically decreased β-arrestin recruitment and inflammatory cascades in chronic kidney disease [27]. However, in our study, the combination of a lower dose of losartan and DMX-200 (90 mg/kg/day) did not show any effect in preventing or arresting aortic dilation, and a further increase in the DMX-200 doses would not be translatable to humans [31,32].

This study has some limitations. While we investigated the combined treatment of barbadin and DMX200 with losartan, we did not evaluate their individual effects. Consequently, definitive conclusions regarding the impact of barbadin and DMX200 alone on aortic aneurysm in MFS mice cannot be drawn. However, the current purpose of our study was to show an additive positive effect of these compounds on top of lower losartan doses, as this would have clinical relevance. We addressed this question and observed no significant synergistic effect, at least with the doses tested in this study.

In conclusion, we observed that none of the tested compounds targeting downstream effectors of AT1R signalling exerted a significant effect on the aneurysms in the MFS mouse model *Fbn1^C1041G/+^*^.^ In contrast, a high dose of losartan (50 mg/kg/day), which blocks the AT1R, was able to prevent the development of aortic dilation. This finding could indicate that full AT1R blockade might be required for the inhibition of aneurysm progression in MFS. Such inhibition could be obtained via an additive effect of another compound on top of a lower losartan dose. [10]. Alternatively, the observed effects of the high dose of losartan on aortic diameter may be independent of the AT1R, indicating alternative mechanisms. Indeed, a potential additive effect of treatment with losartan and β-blockers, which primarily target blood pressure and are not AT1R-related, has been suggested [10]. However, due to insufficient statistical power in the studies, conclusive evidence has not been established. Alternatively, another strategy could involve more site-directed losartan-mediated AT1R blockade based on nanoparticles designed to target the aneurysmal aortic area. The latter could potentially provide a more localised and stronger effect of locally delivered losartan. This approach could lead to better outcomes without the accompanying side effects of using high doses.

## 4. Materials and Methods

### 4.1. Mice

*Fbn1*^C1041G/+^ and WT mice on a C57Bl6J background were used in this study. The total number of mice used per treatment group and genotype was N = 10. However, for plot representation, the number of mice included decreased to N = 7 due to animal loss from non-cardiovascular-related causes or the insufficient quality of the echocardiography imaging data. *Fbn1*^C1041G/+^ and WT mice were housed together up to a maximum of 8 mice per cage. All mice had ad libitum access to food and water. Mice were treated for 8 weeks, starting at 4 weeks of age. After the end of the treatment, mice were euthanised via CO_2_ inhalation.

### 4.2. TRV027 Treatment

TRV027 (Trevena Inc., Chesterbrook, PA, USA) was infused on *Fbn1*^C1041G/+^ and WT mice using micro-osmotic pumps (Alzet model 1004, Durect corporation, Cupertino, CA USA) at an infusion rate of 10 µg/kg/min. The pumps were implanted at 4 weeks of age and only changed once, at 8 weeks of age. In the control group, osmotic pumps were implanted with saline solution following the same procedures.

### 4.3. Barbadin and Losartan (25 mg/kg/day) Treatment

*Fbn1*^C1041G/+^ and WT received daily intraperitoneal injections with either barbadin (0.9 mg/kg/day) or DMSO solution (control). Barbadin (2774, Axon Medchem B.V., Groningen, The Netherlands) was first dissolved in DMSO (15408099, Fisher Scientific B.V., Brussel Belgium) at a concentration of 3.33 mg/mL and subsequently diluted in PBS-Tween80 (11590476, Fisher Scientific B.V., Brussel, Belgium and P1754, Merck, Hoeilaart, Belgium) at a dose of 0.9 mg/kg/day, accounting for the weight of each animal. The same procedure was applied for the control group, but DMSO solution was used instead of barbadin. The injections started at 4 weeks and continued until 12 weeks of age. All mice were additionally treated with losartan at a dose of 25 mg/kg/day. Losartan was dissolved in drinking water at a concentration of 0.3g/L, and drinking bottles were replaced once per week.

### 4.4. DMX-200 and Losartan (5 mg/kg/day and 50 mg/kg/day) Treatment

*Fbn1*^C1041G/+^ and WT were treated with a high dose of losartan (50 mg/kg/day) or low dose of losartan (5 mg/kg/day) and DMX-200 (396265, Merck, Hoeilaart, Belgium) (90 mg/kg/day) in drinking water. Losartan was dissolved in drinking water at a concentration of 0.6 g/L to obtain the high dose of losartan and 0.06 g/L for the low dose of losartan. DMX-200 was dissolved at a concentration of 0.36 g/L. The solutions were changed once per week.

### 4.5. Transthoracic Echocardiography

To visualise the aorta, the hairs of unsedated mice were removed with “Veet sensitive skin” cream. Next, mice were weighted (Appendix A) and subjected to echocardiography using a VisualSonics Vevo 2100 imaging system (FUJIFILM VisualSonics, Inc., Toronto, Canada) and a 30 MHz transducer. Both aortic root and ascending aorta (Appendix A) were imaged in B-mode and PLAX view. Measurements of the aortic root were taken in the sinus of Valsalva, while measurements of the ascending aorta were taken in the mid-ascending aorta. Three independent measurements from the maximal internal aortic dimensions were averaged. All data acquisition steps and measurements were performed while blind to genotype and treatment.

### 4.6. Statistics

Data are represented in interquartile range (IQR) boxplots, in which the error bars represent minimum and maximum values, the horizontal bars indicate median values, and the extremities of the boxes indicate interquartile ranges. Comparison of the aortic diameters between multiple groups within each timepoint was performed with a one-way ANOVA analysis, followed by a post hoc analysis with Tukey’s multiple comparisons test (Appendix A). Differences in aortic growth between the two groups were tested with a Mann–Whitney U-Test. A value of *p* ≤ 0.05 was considered statistically significant. Data analysis and plotting were carried out using GraphPad Prism software version 9.0.0.

## Figures and Tables

**Figure 1 ijms-25-05025-f001:**
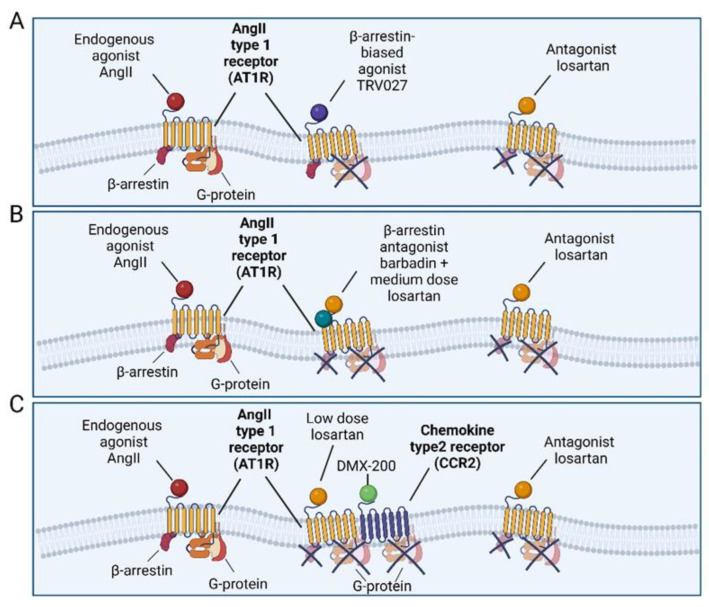
Mechanism of action of the tested compounds. (**A**) TRV027 functions as a β-arrestin-biased ligand, competitively inhibiting G-protein-dependent signalling while promoting β-arrestin recruitment. (**B**) Barbadin selectively blocks β-arrestin. The combination of barbadin with losartan, a AngII antagonist, may result in the potentiated inhibition of AT1R. (**C**) DMX-200 acts as a CCR2 inhibitor. CCR2 couples with AT1R, triggering inflammatory cascades. By combining losartan with DMX-200, AT1R blocking may be further potentiated by also inhibiting subsequent inflammatory pathways. Image created with BioRender.com.

**Figure 2 ijms-25-05025-f002:**
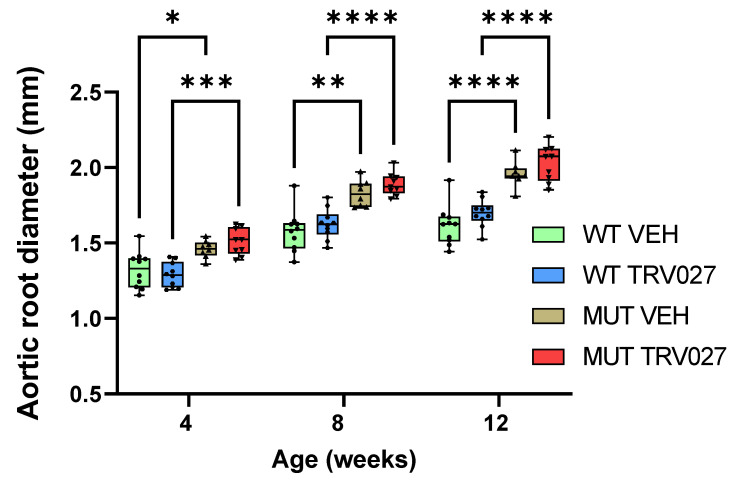
TRV027 does not have any effect on the aortic root diameter of MFS mice. Echocardiography data at 12 weeks show that the aortic root diameter of MFS mice treated with TRV027 during a period of 8 weeks are not significantly different to those treated with vehicle solution. MFS mice from both groups (vehicle and TRV027-treated mice) show a significant dilation before the start of the treatment (4 weeks), during the treatment (8 weeks), and after the end of the treatment (12 weeks). Data are represented as boxplots. Statistical test analysis: One-way ANOVA and Tukey’s post-test per timepoint. **** *p* ≤ 0.0001, *** *p* ≤ 0.001, ** *p* ≤ 0.01 and * *p* ≤ 0.05. VEH: vehicle; WT: wild type; MUT: mutant.

**Figure 3 ijms-25-05025-f003:**
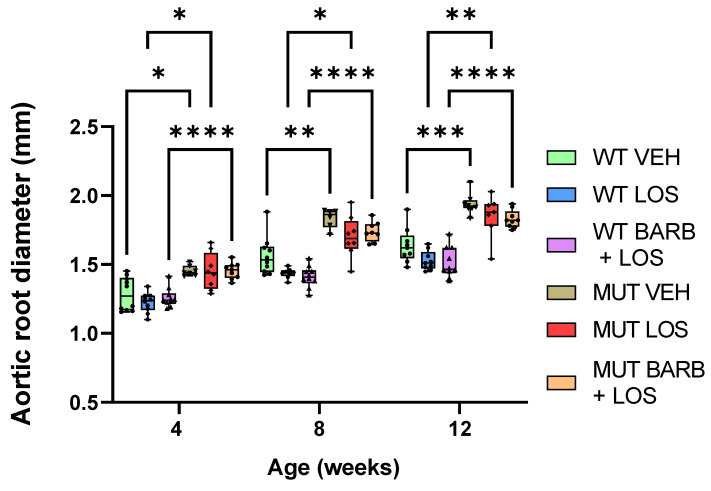
Combined treatment with barbadin and losartan does not show any significant effect on the aortic root diameter of MFS mice compared to losartan alone. Echocardiography data at 12 weeks show that aortic root diameters of MFS mice treated during a period of 8 weeks with barbadin and losartan are not significantly different to those treated with losartan alone. MFS mice from both groups (barbadin and losartan and losartan alone) show a significant dilation compared to the WT controls before the start of the treatment (4 weeks), during the treatment (8 weeks), and after the end of the treatment (12 weeks). A slight decrease in diameter is observed at 8 and 12 weeks for MFS and WT mice of both treated groups compared with the vehicle, but this difference lacks statistical significance. Data are represented as boxplots. Statistical test analysis: One-way ANOVA and Tukey’s post-test per timepoint. **** *p* ≤ 0.0001, *** *p* ≤ 0.001, ** *p* ≤ 0.01 and * *p* ≤ 0.05. VEH: vehicle; LOS: losartan; BARB: barbadin; WT: wild type; MUT: mutant.

**Figure 4 ijms-25-05025-f004:**
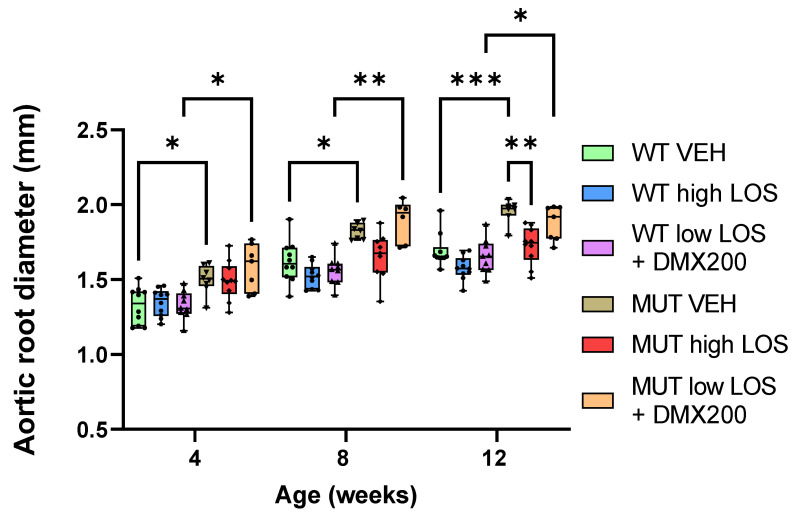
Combined treatment of DMX-200 and low dose of losartan does not show any significant effect on the aortic root diameter of MFS mice compared to both treated and untreated WT groups. Echocardiography data at 12 weeks show that aortic root diameters of MFS mice treated during a period of 8 weeks with DMX-200 and low dose of losartan are significantly different to those in the WT group receiving the same treatment. On the other hand, MFS mice treated with high dose of losartan alone show no difference to the treated WT group during the whole treatment period (8 weeks in total). Similarly, no significant difference is observed between the MFS group treated with high dose of losartan and vehicle WT group at 12 weeks. A slight decrease in aortic diameter is also observable in WT group treated with high dose of losartan when compared to WT vehicle group. Data are represented as boxplots. Statistical test analysis: One-way ANOVA and Tukey post-test per timepoint *** *p* ≤ 0.001, ** *p* ≤ 0.01 and * *p* ≤ 0.05. VEH: vehicle; LOS: losartan; WT: wild type; MUT: mutant.

**Figure 5 ijms-25-05025-f005:**
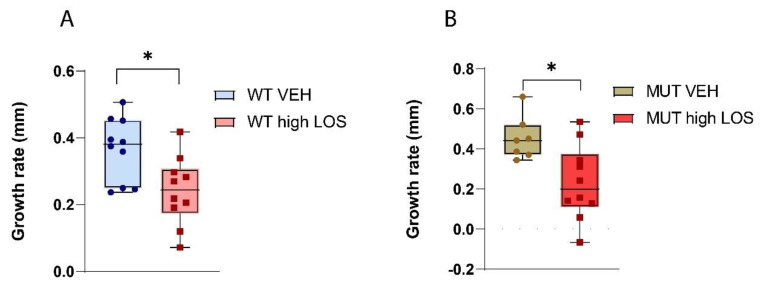
Growth rate of mice upon no treatment and high dose of losartan. The growth rate is represented by the difference in the aortic root diameter between 4 and 12 weeks of age. A decrease in the aortic root diameter of WT (**A**) and MFS (**B**) mice is observed upon treatment with a high dose of losartan when compared to the untreated group. Statistical test analysis: Mann–Whitney U-Test. * *p* ≤ 0.05. LOS: losartan; VEH: vehicle; WT: wild type; MUT: mutant..

## Data Availability

The original contributions presented in this study are included in the article/Appendix A; further inquiries can be directed to the corresponding author. The raw data supporting the conclusions of this article will be made available by the authors on request.

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
