# Peer review of "Investigation of Strategies to Block Downstream Effectors of AT1R-Mediated Signalling to Prevent Aneurysm Formation in Marfan Syndrome"

_ijms, 2024, doi:10.3390/ijms25095025_

Round 1

Reviewer 1 Report

Comments and Suggestions for Authors

In this report Callejon et al investigated the potential beneficial effects of beta-arrestin in combination with low and high doses of the angiotensin II type I blocker, losartan, on the progression of aortic root aneurysm in a well-established mouse model of firbrillin-1 mutation-mediated Marfan syndrome associated aortic root aneurysm. To address the main objectives of the proposed research, the group used losartan at a low dose (25 mg/kg/day) or high dose (50mg/kg/day) alone or in combination with TRV027 that is a beta-arresting-biased AT1R ligand that competitively blocks downstream G-protein-dependent signaling while promoting beta-arrestin recruitment. The rationale for this approach comes from previous studies showing that TRV027 has vasodilatory and cardioprotective effects in laboratory animal models. They also looked at the effects of beta-arrestin inhibition by selective inhibitor Barbadin in combination with losartan on aortic root aneurysm in mice.

The group also looked at the potential beneficial effects of combination of losartan with CCR2 selective inhibitor DMX-200 on aortic root aneurysm progression with the aim of inhibiting potential contribution of inflammatory pathways induced by monocyte chemoattractant protein I (MCP-1) cross talk with AT1R.

The study addresses an important question and is clearly written and structured. The experiments are well executed. However, I have a few concerns and questions that I hope the authors can address before the study is accepted for publication.

Major comments:

1) What was the rationale for including male mice in this study?

2) The authors mentioned that according to their power analysis they needed a sample size of N=10 per group to ensure enough power for the study. However, they also added that due to unexpected death their sample size reduced to 7 per some groups. I am concerned that the reduced power has impacted some of the data presentation and interpretation, specifically in those cases where despite the obvious difference between the means of groups, no significance was detected. On good example is the comparison between Marfan + VEH and MFS + TRV027, which is reported as non-significant. How confident the authors are about this observation?

3) Did the authors measure the blood pressure in experimental mice? Would TRV027 have any vasodilatory effects in Marfan mice?

4) Throughout the report there are references to “aortic root” and “ascending aorta” diameters. Please make it specific what areas exactly were measured. For example: aortic annulus, sinus of Valsalva, sinotubular junction, etc……

5) Figure 3: The animal should have been also treated with barbadin only to assess the effects of Beta Arrestin inhibition only on aortic root aneurysm. This is necessary in order to exclude any potential negative impact of Losartan and barbadin combinational therapy and would have further confirmed the authors’ claim on line 216 that “this suggest losartan alone may be responsible for this effect.”

6) Figure 3: Why did the authors not include a high dose Losartan + barbadin in this part of the study?

7) Figure 4: This data set is not complete without the inclusion of DMX200 only groups. This leaves a big gap in understanding whether inhibition of MCP-1 signal could have any protective or detrimental effects on aortic root aneurysm. This gap is also seen in figure 3 data.

8) Please provide a short description of the current practice in Marfan aneurysm patient management with respect to recommended doses of Losartan in pediatric, adolescent, and adult patients. That helps your reader to put the suggested doses in your study in perspective while interpreting the presented data. The main question here is what is considered a low dose or a high dose in different age groups?

Minor Comments:

1) Please remove all non-significant bars from your bar-graphs and only include the significant p values. In the current crowded format, it is very difficult to read through the data.

2) Try to shorten the introduction if possible. In the current format the introduction is too long. Some of the content can be transferred to the discussion section.  

Reviewer 2 Report

Comments and Suggestions for Authors

The article provides valuable information to science; however, the statistical analysis requires correction.

- The conclusions in the abstract are not supported by specific data (p-value, etc.).

- For the applied statistical tests, it is advisable to calculate appropriate measures of effect size. The p-value alone is definitely not sufficient.

- Given the sample size and other conditions, it is recommended to use the Kruskal-Wallis test and an appropriate post-hoc test like Dunn's.

- Figures 3 and 4 are very unclear; it's difficult to distinguish between elements, the color scheme is not visible, and the boxes and overlaid points are hardly discernible.

- It would be better to present a table with descriptive statistics and relevant effect size measures.

Comments on the Quality of English Language

Minor editing of English language required.

Round 2

Reviewer 1 Report

Comments and Suggestions for Authors

also the authors have addressed a few of my comments and concerns, their lack of desire to scientifically address two major gaps in their study (lack of barbadin only DMX200 only treated groups in figures 3 and 4) makes it difficult to interpret the date. Based on their assumptions only, the authors ignore the complexity of interactions between the two drugs and the importance of potential synergism and antagonism that could impact the combinational treatment. 

Reviewer 2 Report

Comments and Suggestions for Authors

The recommended changes have been implemented.

Comments on the Quality of English Language

Minor editing of English language required.

Author Response

Dear reviewer 2, we wrote the text in English language to the best of our abilities, please let us know where you cant to see minor editing if still needed.

Round 3

Reviewer 1 Report

Comments and Suggestions for Authors

Although the study tries to address an important question, there are gaps in presented data. Unfortunately the validity of data conclusions and interpretation without proper internal controls for com internal therapy is the main concern here.

Author Response

See extensive addtional answer to editor